

**MANUSCRIPT SUBMITTED TO ACP: GMOS-A**
**Title**
**A Smart Nanofibrous Material For Adsorbing and Real-Time Detecting Elemental**
**Mercury In Air**
**Authors**
Antonella Macagnano[2], Viviana Perri[1], Emiliano Zampetti[2], Andrea Bearzotti[2], Fabrizio De
Cesare[4], Francesca Sprovieri[3], Nicola Pirrone[2]
[1]University of Calabria, via Pietro Bucci, Arcavacata, Rende  87036 (CS), Italy; [2]Institute of
Atmospheric Pollution Research-CNR, Via Salaria km 29,300 Montelibretti 00016 (RM), Italy;
[3]Institute of Atmospheric Pollution Research-CNR, Division of Rende, c/o UNICAL-Polifunzionale
Arcavacata, Rende  87036 (CS) Italy; [4]DIBAF-University of Tuscia, Via S. Camillo de Lellis,
01100 Viterbo,  Italy
***Correspondence to:*** *Antonella Macagnano,  a.macagnano@iia.cnr.it; antonella.macagnano@cnr.it*
**Abstract**
The combination of gold affinity for mercury with nanosized frameworks has allowed to design
and fabricate novel kinds of sensors with promising sensing features for environmental
applications. Specifically, conductive sensors based on composite nanofibrous electrospun layers
of titania easily decorated with gold nanoparticles were developed to obtain nanostructured
hybrid materials, capable of entrapping and revealing GEM traces from environment. The
electrical properties of the resulting chemosensors were measured. Few minutes of air sampling
were sufficient to detect the concentration of mercury in the air, in the range between 20-100
ppb, without using traps or gas carriers (LOD ~ 1.5 ppb). Longer measurements allowed the sensor
to detect lower concentrations of GEM. The resulting chemosensors are expected to be low-cost,
very stable (due to the peculiar structure), and requiring low power, low maintenance and simple
equipment to work.

**1 Introduction**
Mercury (Hg) is released into the atmosphere both by human's activities, predominantly fossil fuel
combustion, and naturally, for example, from soil out-gassing, volcanoes and evasion from the sea
(Pirrone et al., 2010; Pacyna et al., 2010). One of the more troublesome questions in recent years





has been to quantify not only the strength of emission sources but also the effects of re-emission of
previously deposited Hg on the overall distribution, concentration and speciation of Hg in the
atmosphere (Hedgecock et al., 2003). The deposition of atmospheric Hg depends on its chemical
speciation, where the term speciation is used to distinguish between the gaseous elemental (GEM)
and gaseous oxidized forms of Hg [GOM and Particle bound mercury (PBM)] and their chemical-
physical characteristics (Lyman et al., 2010; Sprovieri et al., 2016a,b). To be precise, Total gaseous
mercury (TGM) mainly comprises GEM with minor fractions of other volatile species (e.g., HgO,
$HgCl_2$, $HgBr_2$, $CH_3HgCl$, or $(CH_3)_2Hg$). However, in spite of conceptual differences between TGM
and GEM, they have often been used without clear distinction. This was allowable to a degree as
the predominant fraction of TGM (usually in excess of 99%) is often represented by GEM under
normal conditions. GEM is relatively inert under atmospheric conditions, only slightly soluble and
also quite volatile, whereas the oxidized Hg forms found in the atmosphere are both soluble and
involatile, thus they are efficiently scavenged and consequently deposited by liquid atmospheric
water, such as rain and fog droplets, but also deliquesced aerosol particles. The dispersion of GEM
on global scale therefore, depends on the rate of its oxidation in the atmosphere as this determines
its long atmospheric lifetime (generally >1 year), limiting local emission controls from protecting
all environments. Several international initiatives and programs [i.e., the United Nations
Environment Program (UNEP)] have also made a tremendous effort in identifying and quantifying
Hg pollution across the globe, especially the ''hot-spots'', aimed at reducing risk of exposure to this
neurotoxin pollutant. Policy makers are working toward a worldwide effort for supporting the
constructing an accurate global Hg budget and to model the benefits or consequences of changes in
Hg emissions, for example, as proscribed by the Minamata Convention. Anticipating a global
policy, in 2010 the European Commission began a five-year project called the Global Mercury
Observation System (GMOS, www.gmos.eu) to create a coordinated global network to gaps in
emissions monitoring and in the spatial coverage of environmental observations, mostly in the
tropical regions and Southern Hemisphere, thus adequate for improving models and making policy
recommendations (Sprovieri et al., 2016a,b). To date the GMOS network consists of more 43
monitoring stations worldwide distributed including high altitude and sea level monitoring sites,
and located in climatically diverse regions, including polar areas (Sprovieri et al., 2016a,b). One of
the major outcomes of GMOS has been an interoperable e-infrastructure developed following the
Group on Earth Observations (GEO) data sharing and interoperability principles which allows us to
provide support to UNEP for the implementation of the Minamata Convention (i.e., Article 22).
GMOS activities are currently part of the GEO strategic plan (2016–2025) within the flagship on
"tracking persistent pollutants". The overall goal of this flagship is to support the development of



GEOSS by fostering research and technological development on new advanced sensors for in situ
and satellite platforms, in order to lower the management costs of long-term monitoring programs
and improve spatial coverage of observations. Since automated measurement methods of Hg require
power, argon gas, and significant operator training, they are difficult to apply for understanding Hg
air concentrations and deposition across broad regional and global scales. Therefore, the lack of an
inexpensive, stand-alone, low power, low-maintenance sensor is a primary technical issue to be
solve for the sustainability of a global network such as GMOS. Previous research highlighted that
Hg-concentration levels in air vary greatly across different environmental locations, remote as the
Polar Regions, background or rural, and urban locations with an average range between 1.5 ngm$^{-3}$
(GEM) and 1 pgm$^{-3}$ (GOM and PBM), depending on the speciation. Hence, for the determination of
atmospheric Hg also at such low levels, sampling and analytical methods should be sensitive
enough to quantify the concentration profiles of diverse Hg species in each respective
environmental setting to better understand their environmental behavior and patterns. Fortunately,
many advances made in analytical methodologies have made it possible to study atmospheric Hg in
different environmental locations. However, several limitations and difficulties have still
experienced in Hg analysis, as most methods cannot yet directly or accurately determine minor Hg
species (Gustin et al., 2013). Hence, efforts should be continued to secure further the reliability, the
traceability, and the accuracy of Hg levels measured in air. Current air monitors are amply sensitive
to detect the global background but are costly, complicated configuration, electricity requirements
and high maintenance. This limits the scientific research community's ability to long-term measure
atmospheric Hg concentrations worldwide. Sampling and analysis of atmospheric Hg is made most
commonly as GEM/TGM because of their greater abundance, even if both manual and automatic
methods have been currently developed for different Hg forms to suit the measurement and
monitoring application. The most common sampling method employed relies on adsorption on gold
amalgam and then, either directly or indirectly, through a stepwise process of thermal desorption
and final detection [usually by cold-fiber atomic absorption spectroscopy (CVAAS) or cold-fiber
atomic fluorescence spectroscopy (CVAFS)]. However, our knowledge presents currently several
gaps to be solved. Firstly, The atmospheric chemistry of Hg remains poorly understood, especially
the oxidation pathways by which GEM is converted to GOM, the reduction pathway which converts
GOM back to GEM, and the gas-particle partitioning. This is partially due to the need for
identification of the chemical forms of oxidized Hg in the atmosphere and methods to measure
these compounds individually. In addition, the limitations and potential interferences with our
current measurement methods have not been adequately investigated, thus alternate methods to
measure atmospheric Hg are needed. Given the uncertainty and restrictions associated with





automated and/or semi-automated Hg measurements (Gustin et al., 2013; Pirrone et al., 2013), and
above all, responding to the technical needs of an expanding Hg global observation network, we
developed a reliable, sensitive, and inexpensive surface for atmospheric Hg detection. In particular,
we investigated and demonstrated the utility of composite nanofibrous electrospun layers of titania
decorated with gold nanoparticles (AuNPs) to obtain nanostructured materials capable of adsorbing
GEM as a useful alternative system for making regional and global estimates of air Hg
concentrations. Methods and new sampling systems previously developed, such as passive
samplers, have been used to understand long-term global distribution of persistent organic
pollutants (POPs) (Harner et al., 2003; Pozo et al., 2004). Other passive samplers for both TGM and
GOM collection on the basis of diffusion have been constructed using a variety of synthetic
materials (i.e., gold and silver surfaces, and sulfate-impregnated carbon) and housings (Lyman et
al., 2010; Gustin et a., 2011; Zhang et al., 2012; Huang et al., 2014). However, because of the
differences in design of passive samplers, ambient air Hg concentrations quantified by various
samplers may not be comparable. In addition, sampling rates (SRs) using the same passive samplers
may depend on environmental conditions and atmospheric chemistry at each site. Moreover, it has
been also highlighted that the performance of passive samplers may be influenced by
meteorological factors (e.g., T °C, RH, wind speed) therefore inducing bias for the result of passive
sampling (Plaisance et al.,. 2004; Sderstrm et al., 2004). On the other hand, incentive for developing
simple and cost-effective samplers that are capable of monitoring over an extended period and
require no technical expertise for deployment of these systems also at remote locations is now
obvious. In this work we describe an alternative approach adopted in the place of conventional ones
demonstrating that the combination of gold affinity for Hg with a nanoscale sized framework of
titania provided the chance to create promising sensors for environmental monitoring in real time,
characterised by high sensitivity to the analyte. The novel sensor is a relatively simple and low cost
method for measurement of the most abundant Hg form in ambient air (TGM/GEM) due to reusable
parts and simple deployment steps. Further, we have evaluated the applicability of this
measurement technique with respect to real environmental conditions highlighting future directions
of research on airborne Hg determination. The TGM/GEM sensor surface described here could be
deployed in a global network such as GMOS; a permanent network of ground based monitoring
sites and observations of Hg and/or related species on a global scale and with remote sensors would
in fact be highly desirable. These data are needed to test and validate model processes and
predictions, understand the source-receptor relationships, understand long-term changes in the
global Hg cycle, and at least, would help policy makers to set regulations for different areas.  The
sensor features   are related to the nanofibrous scaffold of titania capable of growing up gold



nanostructures by photocatalysis, tunable in size and shape. Such a nanostructured layer, fabricated
by electrospinning technology, firstly improves sensor features with respect to those of compact
films, by enhancing the global number of binding sites of analyte-sensor and reducing some bulk
drawbacks. Secondly, the combination of metal oxides and metal nanostructures, improves the
sensitivity, allows sensor to work at room temperature, tunes selectivity towards different gas
species by adjusting the surface to volume ratio of nanosized structures and affect sensor lifetime.
Morphological, optical, electrical aspects and sensing measurements of fibers of GEM in air have
been reported and discussed. When designed, the resulting Hg ad-absorbent material was expected
to be suitable for novel Hg sensors fabrication, since a similar nanofibrous scaffold doped with
AuNPs was described in literature as filtering systems capable of adsorbing and removal Hg fiber
from the environment with an efficiency of about 100% (Y Yuan et al., 2012). In fact, in previous
works (Macagnano et al., submitted, Macagnano et al., 2015a) the authors reported the ultra-high
sensitivity of the sensor, capable to detect up to dozens ppt, despite of a long time necessary to
reveal the analyte at these concentrations, in air. In this work the chance to apply the sensor in
polluted sites and in real time has been presented and described.

**2 Materials and methods**
*2.1. Chemicals*
All chemicals were purchased from Sigma-Aldrich and used without further purification:
polyvinylpyrrolidone (PVP, Mn 1,300,000), titanium isopropoxide (TiiP, 99.999%), gold(III)
chloride hydrate (HAuCl$_4$, 99.999%), anhydrous ethanol (EtOH$_a$) and glacial acetic acid (AcAc$_g$).
Ultrapure water (5·5 10$^{-8}$ S cm$^{-1}$) was produced by MilliQ-EMD Millipore.
*2.2. Electrospinning technology*
Electrospinning is a widely used technique for the electrostatic production of nanofibers, during
which an electric field is used to make polymer fibers with diameters ranging from 2 nm to some
micrometres from polymer solutions (or melts). It is currently the most economic, versatile, and
efficient technology to fabricate highly porous membranes made of nano and/or micro fibers also
for sensors (Macagnano et al., 2015b). It is based on the application of a high voltage difference
between a spinneret ejecting a polymeric solution and a grounded collector. The jet of solution is
accelerated and stretched by the external electric field while travelling towards the collector, leading
to the creation of continuous solid fibers as the solvent efiberates. The electrospinning apparatus
used in the present study (designed and assembled in CNR laboratories) comprised a home-made
clean box equipped with temperature and humidity sensors, a syringe pump (KDS 200, KD
Scientific) and a grounded rotating cylindrical collector (45 mm diameter), a high voltage oscillator



(100 V) driving a high voltage (ranging from 1 to 50 kV) and a high power AC-DC (alternative
current-direct current) converter. Electrospinning solution ($7.877 \times 10^{-5}$ M), was prepared by
dissolving PVP in $EtOH_a$ and stirring (2 hours). A 2 ml aliquot of 1:4 (w/v) solution of TiiP solved
in 1:1 (v/v) mixture of $AcAc_g$ and $EtOH_a$ was freshly prepared and added to 2.5 ml PVP solution
under stirring in order to obtain a 1.95 (w/w) TiiP/PVP final ratio. Both mixtures were prepared in a
glove box under low humidity rate (<7% RH). The syringe filled with the TiiP/PVP solution and
housed in the syringe pump, was connected to a positive DC-voltage (6 kV), and set to a 15 cm far
grounded rotating collector. The substrates were fixed through suitable holders onto the collector
(600 rpm, 21 °C and 35% RH) and processed (feed rate 150 ml $h^{-1}$) for 20 min to obtain scaffolds
for sensors. After deposition, $PVP/TiO_2$ composite nanofibers were left for some hours at room
temperature to undergo fully self-hydrolysis of TiiP [Li et al., 2003]. And then annealed under
oxygen atmosphere (muffle furnace) using a thermal ramp from room temperature up to 550 °C (1
°C $min^{-1}$, 4 h dwell time) in order to remove PVP and crystallize the metal oxide (*anatase*).
*2.3 Transducers: interdigitated electrodes*
The transducer adopted in the present work to convert the physico-chemical interactions of analytes
with the different polymer fibers in an electrical signal was an interdigitated electrode (IDE) [Bakir
et al,1973; James et al., 2013].  Specifically, the transducer consisted of 40 pairs of electrodes (150
nm in electrode thickness, 20 μm in gap and electrode width and 5620 μm in length) was
manufactured in CNR laboratories through a standard photolithographic process (lift-off
procedure), then followed by Ti sputtering and Pt efiberation, suitable to generate the electrodes of
the size reported above, on a 4 in. oxidised silicon wafer. After electrospinning deposition all the
electrical signals of the resulting chemoresistors were recorded by an electrometer (Keithley 6517
Electrometer).
*2.4 Titania nanofibers*
Upon calcination, the diameters of fibers extraordinarily shrunk: mean diameters of fibers were
estimated through image analyses to be approximately within the range of 60–80 nm. Specifically,
the resulting fibers appeared fine and rough at surface, with a fairly homogeneous fabric. The
absence of beads and the good quality of the long and continuous fibers was confirmed through
SEM micrographs. A highly porous and dense network of nanofibers covering the electrodes was
observed, showing interconnected void volumes (porosity) and high surface-to-volume ratios
(specific surface area). Zampetti et al., (2013) reported that such a fibrous layer showed a 99% of
pores having an area less than 10 $\mu m^2$, with more than 80% pores being <1 $mm^2$.
*2.5 AuNPs/TiO2NFs photocatalytic decoration*





Exploiting the photocatalytic properties of $TiO_2$, gold nanoparticles were selectively grown, under
UV-light irradiation, on the electrospun titania nanofibers through the photoreduction of $HAuCl_4$ in
the presence of an organic capping reagent (PVP). Thus the resulting fibrous scaffolds were dipped
into an aqueous solution containing $HAuCl_4$ and PVP ($1.5 \cdot 10^{-3}$M and 0.1M respectively) and
exposed to UV light irradiation for specified intervals (UV lamp (365 nm) (Helios, Italquartz).
Depending on the gold nanoparticles size that were forming in photocatalysis, the dip-solution
changed from light yellow to purple. Samples were rinsed extensively with water and then air-dried.
Before morphological, electrical and sensing measurements, samples were heated at 450 °C per 1 h
to eliminate the PVP traces. Morphological characterization were provided by scanning electron
microscopy (SEM) (Jeol, JSM 5200, 20 keV) with pictures captured at 5 kV accelerating voltage.
AFM (atomic force microscopy) micrographs were taken  in tapping mode using 190Al-G tips, 190
kHz, 48N/m (Nanosurf FlexAFM). TEM (C-TEM, control transmission electron microscopy)
micrographs were performed at 200 keV with an analytical double tilt probe. TEM specimen were
prepared by gently scraping at first the TiO2 nanofibrous layer electrospun onto the silicon support
and then collecting the nanofibers, through adhesion upon contact with holy carbon thin film. UV-
Vis spectra were provided by Spectrophotometer UV-2600 (Shimadzu), analysing quartz slices
coated with nanofibers. These substrates were able to collect fibers by electrospinning (20 min), and
then were subjected to calcination according to the described above procedure, and then UV
irradiation in the aqueous solution. The fibrous layer stayed stuck to the substrate if the thickness
was thin enough. Longer depositions caused curling of fibers during the calcination process.
*2.5 Measurement set-up*
The sensor was placed in a suitable PTFE-made measurement chamber (0.7 ml volume) connected
to an electrometer (Keithley 6517 Electrometer) capable of measuring the current flowing through
the IDE, when a fixed potential was applied to it, and to send data to a PC. Dynamic measurements
were carried out at room temperature both using: (i) 4 channel MKS 247 managing four MKS mass
flow controllers (MFC), set in the range 0–200 sccm and (ii) Environics S4000 (Environics, Inc.)
flow controller, containing three MFCs supplying different flow rates (up to 500, 250 and 25 sccm,
respectively), managed by its own software. Pure air (5.0) (Praxair–Rivoira, Italy) was used as gas
carrier. A homemade PTFE (polytetrafluoroethylene) permeation tube filled with a suitable amount
of $Hg^0$ was included within such a delivery system to get set dilutions of Hg-saturated vapours. The
tube was immersed in a thermostatically controlled bath, thus the desired $Hg^0$ concentration
delivered to the sensor was achieved by both tuning the temperature of the permeation tube and the
dilution flow. The $Hg^0$ concentration was checked by Tekran®2537A analyser. Responses were
calculated as $\Delta I/I_0$, where $\Delta I$ was the current variation and $I_0$ was the current when synthetic pure



dry air was flowed. Sensor was restored after a quick thermal shot at 450°C under flow of pure air
(450°C).

**3 Results and Discussion**
Nonwoven mats made of PVP and amorphous TiO2 were obtained by the combination of
electrospinning and sol-gel techniques (Fig. 1). The deposition occurred for 20 min on oxidised
silicon wafers and IDEs, properly fixed on the surface of a conducting rotating collector to form
nanofibrous layers characterized by high surface areas and relatively small pore sizes (Zampetti et
al.,2013). By changing the deposition time, both thickness and consistence of the mats changed: one
hour deposition provided the formation of a thicker white and soft fabric easily peeled off (Fig.1),
hygroscopic and soluble in both water and polar solvents; a 20 min deposition generated a fibrous
film adhering to substrates, too thin to be weeded. The calcination process caused a complete
degradation of PVP with formation of crystalline $TiO_2$ (*anatase*) and a significant shrinkage of
fibers dimension (60-80 nm diameter, 5-40 nm grain sixe). Exploiting the photocatalytic properties
of titania (*anatase*), a tunable decoration of fibers with gold nanoparticle could be achieved by
dipping the fibrous mats in a proper aqueous solution ($HAuCl_4$, PVP) under UV light irradiation (Li
et al, 2004; Macagnano et al., 2015). The photocatalytic reaction was proved by the color of the
solution (red purple from light yellow) (Fig.1). Changing both UV irradiation exposure time and
PVP concentration, as capping reagent, morphology, size and density of gold nanoparticles could be
tuned [Macagnano et al., submitted**]**.

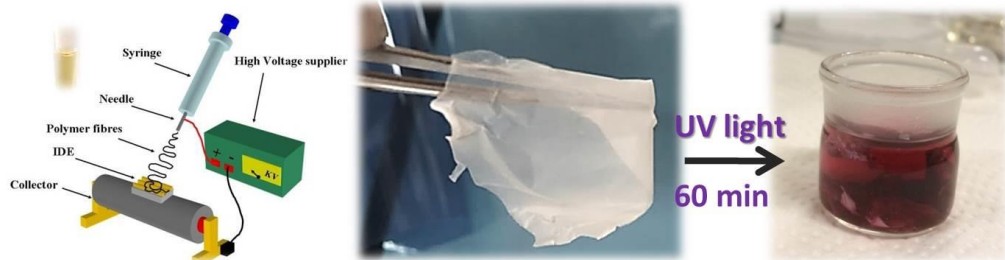

Figure 1. Sketch of an electrospinning set-up comprising a syringe and a grounded rotating cylinder collector where the samples take place for their coverage (*left*);  a piece of a nanofibrous fabric of TiiP/PVP peeled from the substrate after 1 hour of electrospinning deposition (*centre*) a red-purple aqueous solution of $HAuCl_4$/PVP after UV-light irradiation treatment holding a piece of the nanofibrous fabric of $TiO_2$ (*anatase*) obtained after TiiP/PVP annealing (*right*)


In the present work, among a series of differently coated fibrous layers, only the fibrous
nanocomposites that were conductive at room temperature were selected and then their electrical
and sensing features investigated. The controlled gold deposition was due to the photo-excited
electrons on the surface of $TiO_2$ nanofibers that were able to reduce the gold ions thus inducing gold
deposition (Fig. 2, *the sketch*). The capping reagent war responsible of the shape of the particles.





The surfaces of nanofibers observed in SEM micrographs (Fig.2, *right*) appeared densely decorated
with globular nanoparticles. In C-TEM image (Fig.2, *inset*) the gold nanoparticles appeared darker
and spherical or quasi-spherical. The single particles size were ranging between 2 and 20 nm, with a
7.8±3 nm average diameter [Macagnano et al., submitted]. Gold nanoparticles grew directly onto
the nanofibers and their adhesion appeared relatively strong (despite due to van der Waals forces),
since they both resisted to water rinsing and fibers scratching for TEM analyses.

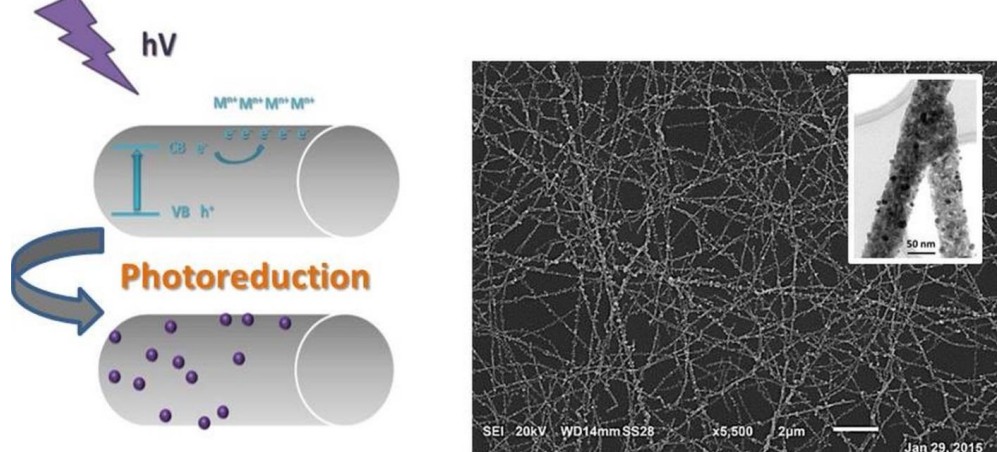

Figure 2. A sketch of the photocatalytic process occurring on the fibers surface (*left*); SEM picture of a dense nanofibrous network of AuNPs/TiO$_2$ coating a silicon wafer (*right*); a C-TEM micrograph of fibers finely decorated with gold nanoparticles (*the darkest ones*) fixed without using any additional linker (*inset*).


After photocatalytic process, the white porous mat became purple-violet. As seen in the spectrum of
the AuNP/TiO$_2$ system, a characteristic absorbance band appeared at around 543 nm, which
corresponded to the surface plasmon resonance (SPR) of the AuNPs (Sun et al, 2003). A red
shifting and broadening of the absorbance band was observed with the increasing in AuNP size and
fiber loading, respectively (data not shown). The colour is strictly depending on the size of the
nanoparticles, and then their agglomeration at the solid state. According to Bui et al. (2007), such a
band broadening phenomenon is due to the electric dipole–dipole interactions and coupling
occurring between the plasmons of neighbouring particles, whereas nanoparticle agglomeration
phenomena occurred.



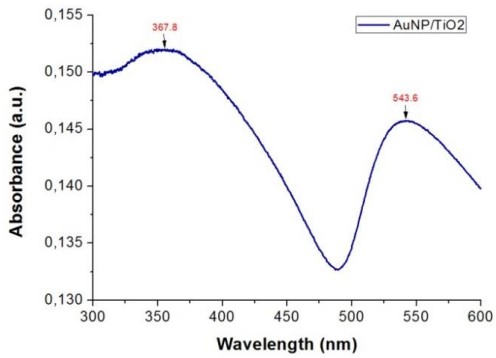

Figure 3. UV-Vis spectrum of a titania nanofibrous network
after gold decoration ($TiO_2$: 367.8 nm; Au NPs: 543.6 nm)


Due to these features, UV-Vis absorption spectroscopy has been used in literature as a technique to
reveal the changes in size, shape and aggregation of metal nanoparticles in liquid suspension after
exposure to heavy metals, as $Hg^0$ (Morris et al., 2002). Both blue-shifted wavelength and its extent
were proportional to the amount of $Hg^0$ that entered the liquid suspension. Similarly, when the gold
decorated nanofibers of titania, collected on a quartz slice, were exposed to $Hg^0$ vapours (2 ppm) in
air for 15 min, a significant blue shifting was reported (~ 3 nm) (Fig. 4) due to the atomic
adsorption of GEM on the surface. The nanoparticles could be regenerated by heating the sample at
550°C for 3 minutes to remove $Hg^0$. The recovery of the AuNPs was stated by the achieving of the
original values of wavelength. The regeneration could be done for dozen times without any
deterioration. Similarly, in chemoresistors, the $TiO_2$ nanofibrous layers attached to the substrates
(Fig. 5), changed colour after photocatalytic treatment.

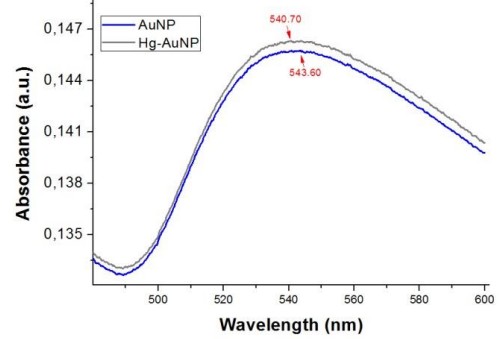

Figure 4. UV-Vis spectra of AuNPs/$TiO_2$nanofibers before
(*blue*) and after a 15 min exposure to 2 ppm of $Hg^0$ (*gray*)




The IDE layout (Fig.5) show a set of interdigitated electrodes which occupies an area
approximately 3x5 mm, completely coated with the sensitive fibers, and two bonding pads (2x2
mm) that will be connected to the electrometer (DC voltage). Such a planar interdigitated electrode
configuration is the most commonly used for conductometric sensing applications.

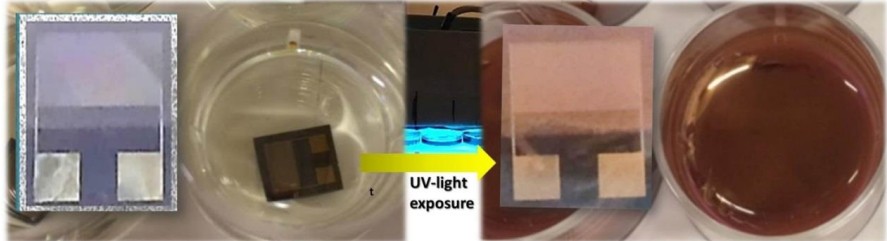

Figure 5. Chemosensor fabrication and final structure: IDE dipped (*left*) and exposed to UV-light (*right*) for gold decoration


Figure 6 depicts a Current–Voltage (I-V) curve of a chemosensor, under synthetic dry air. However
curve shape was unaltered when air or nitrogen were flushed over the fibers (Macagnano et al.,
2015), suggesting that oxygen concentrations poorly affected the electrical properties of such a
chemoresistor. The resistance value of IDE coated with $TiO_2$ nanofibers before photocatalysis,
resulted to be too high at room temperature to contribute straight to the final current. The resulting
linear shape (Ohmic behaviour) within the selected voltage range (−3V to +3V) showed a constant
resistance value for the sensor. The very low value of resistance ($\sim$1.2 k$\Omega$) provided the chance to
work at low voltage, with consequent effects on the energy consumption as well as lifetime of the
material. Moreover, the linearity of I–V curve let us suppose that the sensing scaffold had a good
adhesion to the metal electrodes. The electron conductivity has been described by the percolation
model (Macagnano et al, submitted; Muller et al., 2003) since the titania at room temperature was
supposed like an insulating organic matrix. When it is metal doped, the electron conductivity is
ruled by thermally activated electron tunneling from one metal island (gold nanoparticles) to the
other. However, the conductivity of the nanocomposite is lower than that of pure metal (gold)
because the electron mean free path is greatly reduced due to the presence of the dielectric (the
titania crystals). The electrical features, such as the reproducibility of the fabrication process, of this
conductive device have been previously investigated by the authors (Macagnano et al, submitted;
Macagnano et al., 2015), showing encouraging results for the development of a low cost sensor for
mercury detection. However, in spite of the high sensitivity (LOD: 2ppt) of the sensor, too long
response time was necessary to detect traces of $Hg^0$, when compared to the current monitoring
instrumentations (Ghaedi et al., 2006; Sanchez-Rodas et al., 2010; Ferrua et al., 2007). Extremely





encouraging resulted if compared to other sensors currently involved in detecting mercury in air
(Drelich et al., 2008; Kabir et al., 2015; Sabri et al., 2009; Mohibul Kabir et al., 2015; Raffa et al.,
2006; James et al., 2012-2013; Chemnasiri et al., 2012; Sabri et al., 2011; Keebaugh et al., 2007;
Crosby, 2013; McNicholas et al., 2011). The long time in response was supposed to be in part due
to the layout of the measuring system, since the sensor was housed in a quartz bottle of 100 mL
volume. In fact, an additional time was expected to be caused by the adsorption of the $Hg^0$ traces
from the surrounding environment (measuring chamber) up to achieve a sufficient number of
mercury atoms adsorbed on the surface sensor to be electrically revealed. In the present study the
measuring chamber was designed in order to reduce the volume (0.7 mL) and to expose the fibers to
the gas entry (Fig. 7). Such a measuring layout was designed to allow the fibrous network to be
exposed to the mercury atoms as delivered into the sensor chamber.

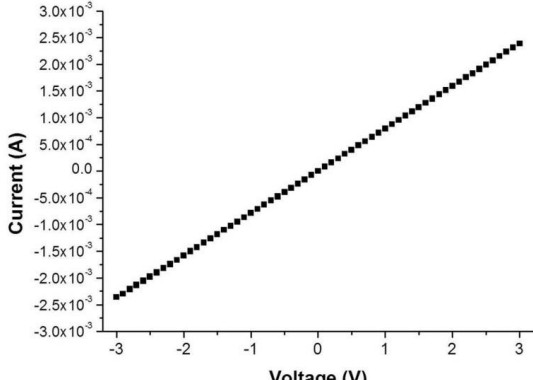

Figure 6. Chemosensor current-voltage curve


Sensing measurements, i.e. current (or resistance) changes, were provided in continuous. The
sensor measurements, that were the electrical signals reported when interaction between the sensing
layers and the analytes were happening, resulted in a change of the whole current (or resistance, i.e.
I =V/R) according to Ohm's law. The sensor was exposed to a flow of $Hg^0$ in air with a
concentration of 800 ppb for 1 min (Fig. 7, *right*), and then air was used to clean the sensor surface.
A rapid decrease in current was recorded ($1.056 \cdot 10^{-7}$ A·s$^{-1}$) when $Hg^0$ entered the measuring
chamber. The current curve trend slightly changed when clean air was flowed, stabilizing at about
the current values reached for $Hg^0$ adsorption. Due to the strong affinity between Au and $Hg^0$, a 3
min-thermal treatment was necessary to remove mercury from layer and get the same starting
current value.





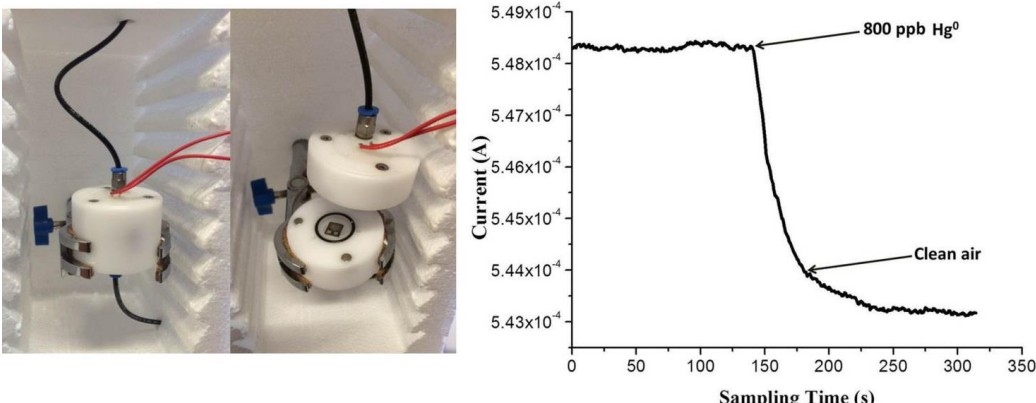

Figure 7. Homemade measurement chamber to house the chemosensor for laboratory experiments (left); plot depicting the transient response curve to 800 ppb Hg$^0$ (V=0.3 V)



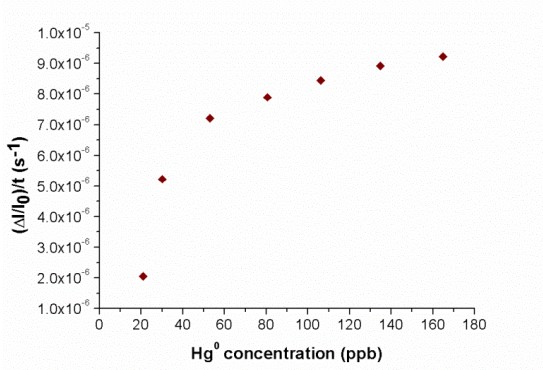

Figure 8. The normalized sensor response rate to the increasing concentration of vapour elemental mercury


Figure 8 depicts the normalized sensor response rate, i.e. the normalized current change per second,
toward the increasing concentration of GEM (ranging between 20 and 160 ppb). The resulting
logarithmic curve describes how the Hg$^0$ concentration affects the response time: small variations of
Hg$^0$ concentration up to 80 ppb are able to deeply change the response rate, on the contrary higher
concentration seem to affect only slightly this sensing feature. Since a strong relationship is
recorded between the concentration and the response time when the content of mercury in
environment is low, is possible to correlate the slope of the transient responses within the first
minutes of the sensor response to definite concentrations of Hg$^0$ in air. Figure 9 depicts the linear
fitting of 10 min-sensor responses when increasing concentrations of mercury were flowed onto the
sensor. Related data were reported in Table 1.





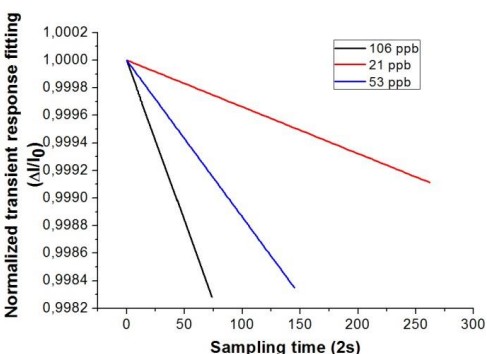

Figure 9. Linear fitting of the normalized sensor response within the first ten minutes




**Table 1. Linear fitting parameters of 10 min-sensor responses to 21 ppb≤ [Hg$^0$] ≤106ppb**

| ppb | $(\Delta I/I_0)s^{-1}$ | SE($\pm$) | R$^2$ |
|-----|------------|-----------|-------|
| 21 | -7.12602E-10 | 1.75521E-11 | 0.86 |
| 33 | -1.50647E-9 | 1.05521E-10 | 0.91 |
| 39 | -1.78067E-9 | 1.02615E-10 | 0.91 |
| 40 | -1.85901E-9 | 1.01833E-10 | 0,92 |
| 53 | -2.44657E-9 | 4.24993E-11 | 0.91 |
| 70 | -3.19082E-9 | 2.55882E-11 | 0.93 |
| 106 | -4.83599E-9 | 2.67462E-10 | 0.88 |


A linear relationships has been reported between the response rate and the concentration of Hg,
according to the following equation (1):

363        (1)    $y = (-4.56226E^{-11}) \cdot [Hg^0], \ [Hg^0] < 100 \ ppb; \ SE: \pm 1.504E^{-12}; \ R^2 = 0.99675$






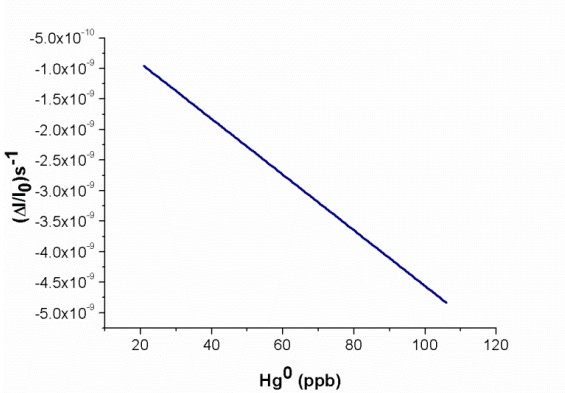

Figure 10. Linear relationships between the normalized response time and the $Hg^0$ concentration, within the range of 20 and 100 ppb.


Therefore when the concentration of Hg increased, the response curve slope changed too linearly,
allowing a limit of detection of about 1 ppb, when the sensor is exposed to air polluted with $Hg^0$ per
10 min. For what concerns main interfering compounds, since at room temperature and in dark
condition the measured current is supposed to be due to AuNPs decorating titania fibers, only
chemical compounds interacting with gold are expected to be mostly responsible of the current
changes (i.e. halides and sulphides). Thus in a blend of other chemicals, this sensor has been
designed as a pretty selective sensor, being able to greatly decrease the environmental disturbances
allowing the investigator/manufacturer to design and then fabricate easier strategies to prevent
contaminations from environment (selective filtering systems or coatings). Among common
potential contaminants authors investigated previously water vapour influence (%RH) reporting no-
effects on the electrical signals (Macagnano et al., 2015).

**4 Conclusions**
The adopted sensing strategy has focused on the strong affinity of mercury to gold combined to
the nanostructures properties. Exploiting the photocatalytic properties of electrospun titania
nanofibers, a novel conductometric sensor has been designed and fabricated to detect GEM in air.
Electrospinning technology has been used successfully to create a 3D-framework of titania
covering the electrode sensing area of the properly designed chemoresistors (IDEs). AuNPs have
been grown on $TiO_2$ nanofibers exploiting the photocatalytic properties. Such a sensor was able to
work at room temperature and was highly sensitive to $Hg^0$. Since it is composed of titania and
gold, it sounds to be robust and resistant to common solvents and VOCs commonly in the air. The



short thermal treatments (450°C, 3min), necessary to desorb mercury from AuNPs, didn't seem to
affect the lifetime of the device. Depending on the strategy of sampling, a sensing device based on
such a chemosensor, could be designed for real applications, specifically for real time monitoring
of polluted sites. Few minutes of sampling of air are sufficient to quantify the concentration of
mercury in the air, in the range between 20-100 ppb (LOD: 1 ppb), without using traps or gas
carriers. However further investigations are necessary also to assess the effects of physical
parameters of the environment, such as temperature fluctuations and UV-light, as well as
chemical ones, such as volatile organic compounds and gas (like halides and sulphides) which are
well known interfering of the adsorption process of the $Hg^0$ on gold.

**Acknowledgments**
The activity is part of the International UNEP-Mercury Programme (UNEP-Mercury Air Transport
and Fate Research (UNEP-MFTP) within the framework Global Mercury Observation System,
funded by EC as part of EC FP7. Furthermore, authors gratefully thank Mr. Giulio Esposito and Mr.
A. Capocecera for their support in the use of laboratory instrumentations.

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
