# Peer review of "MANUSCRIPT SUBMITTED TO ACP: GMOS-A"

_Atmospheric Chemistry and Physics, 2016_

## Referee Comment (RC1) · Anonymous Referee #1 · 14 Jan 2017

A review of the manuscript acp-2016-1077

Journal: ACP Title: A smart nanofibrous material for adsorbing and real-time detecting elemental mercury in air Author(s): Antonella Macagnano, et al. MS No.: acp-2016-1077 MS Type: Research article Special Issue: Global Mercury Observation System – Atmosphere (GMOS-A)

General comments The task of the atmospheric mercury monitoring network developing is very important for understanding the scale of emission, regional and global transfer, and deposition of this environmental pollutant. Existing system of background monitoring is based on AFS and AAS instrumental observations requiring high investments for set up of any new monitoring point. That's why creation of new low-cost, hands-

off measurement systems is an imperative topical issue. The manuscript presents the results of the development of a novel sensor for air mercury measurements based on original manufacturing technology. The title reflects the contents of the paper, the main results are outlined in the Abstract. Introduction gives a comprehensive, 3.5 pages, review of the atmospheric mercury, speciation, mercury transfer, existing publications on the mercury sensors development. Principal part of the manuscript is devoted to description of a new sensor design and manufacturing technology, study of the absorptivity of the new material exposed to mercury vapour. Sensitivity of the new sensors is far not sufficient for the task of mercury monitoring declared in Introduction (see Specific comments, 1). No data on mercury measurement in ambient air, selectivity, and possible interferences are presented.

Specific comments 1) Commonly, in regulatory and scientific literature, the weight concentration units are used (ng/m3, pg/m3) for atmospheric mercury and mercury speciation. The authors use these units in Introduction, but different units: ppm, ppb, ppt in parts describing sensors. It is not explained what these units are related to: volume, mass, or number of molecules? We suppose the volume units are used. It is inconvenient, because a comparison with conventional concentration units requires recalculation to the normal condition (P, T). Besides, the "tiny" ppb values can create a false impression of enabling background Hg concentration measurements. For example, mercury concentration of 1 ng/m3 is about 0.1 ppt (vol). It turns the achieved "low" detection limit (LOD) and measurement range of "20 -100 ppb (LOD 1,5 ppb)" (see lines 23-24) to a quite high figures in conventional units: 200,000 – 1,000,000 ng/m3 (LOD = 15,000 ng/m3). For comparison, the concentration of saturated Hg vapour is 12,000,000 ng/m3 at 18 oC. Such sensitivity gives no possibility to monitor mercury in ambient air, as the LOD is 10,000 times larger than the average background mercury concentration of 1.5 ng/m3 (see line 75).

2) How the LOD value was determined and at what exposure time? There is no description in the text. Different figures of LOD are presented in Abstract and Conclusion

(compare lines 23-24 and 390-391).

3) Lines 69- 70 "Since automated measurement methods of Hg require power, argon gas, and significant operator training, they are difficult to apply. . ." Argon is not required for AAS systems.

4) Lines 105-107 " decorated with gold nanoparticles (AuNPs) to obtain nanostructured materials capable of adsorbing GEM as a useful alternative system for making regional and global estimates of air Hg concentrations". Gold adsorbs not only GEM.

5) Lines 116-118 This is true that: "sampling rates (SRs) using the same passive samplers may depend on environmental conditions and atmospheric chemistry at each site. Moreover, it has been also highlighted that the performance of passive samplers may be influenced by meteorological factors (e.g., T °C, RH, wind speed) therefore inducing bias for the result of passive sampling (Plaisance et al.,. 2004; Sderstrm et al., 2004)." However, there is no evidence that the developed samplers are independent of meteorological factors (e.g. humidity) and interference of trace gases and aerosols, especially under long exposure time.

6) Lines 121-124 "In this work we describe an alternative approach adopted in the place of conventional ones demonstrating that the combination of gold affinity for Hg with a nanoscale sized framework of titania provided the chance to create promising sensors for environmental monitoring in real time, characterised by high sensitivity to the analyte." See note (1). Besides, the technology of real-time monitoring is not described.

7) Lines 128-131. "The TGM/GEM sensor surface described here could be deployed in a global network such as GMOS; a permanent network of ground based monitoring sites and observations of Hg and/or related species on a global scale and with remote sensors would in fact be highly desirable". Again, see note (1). The achieved sensitivity is insufficient for background measurement at the level of few ng/m3 provided by GMOS stations, and for speciation study at the level of pg/m3.

8) Lines 333-338. For better understanding of sensing measurements: "The sensor was exposed to a flow of Hg0 336 in air with a concentration of 800 ppb for 1 min (Fig. 7, right)" In terms of standard units, we see that the sensor was exposed to almost saturated Hg vapour at 8 mg/m3. Thereby it is not clear how such high concentration was measured? Earlier (line 234), it was mentioned, that "The Hg0 concentration was checked by Tekran$^{®}$2537A analyser". Tekran 2537 is not capable of measuring such high concentration.

Technical corrections 1). Numerous misprints: "fibers" instead of "vapour":

Lines 91-91: "cold-fiber atomic absorption spectroscopy . . .. cold-fiber atomic fluorescence spectroscopy"

Line 141: "Morphological, optical, electrical aspects and sensing measurements of fibers of GEM in air"

Line 144: "adsorbing and removal Hg fiber"

The same mistakes in the Reference list:

Lines 412-414 "Ferrua, N., Cerutti, S., Salonia, J. A., Olsina, R. A. and Martinez, L. D., 2007. On-line preconcentration and determination of mercury in biological and environmental samples by cold fiber-atomic absorption spectrometry. J. Hazard. Mater. 141 693–699" In the original Cold Vapour, see: http://www.sciencedirect.com/science/journal/03043894/150

Lines 412-414 "Determination by Cold Fiber Atomic Absorption Spectroscopy. Analytical Letters 39" In the original Cold Vapour, see: http://www.tandfonline.com/doi/full/10.1080/00032710600622167

2) Lines 73 -76 The sentence "Previous research highlighted that Hg-concentration levels in air vary greatly across different environmental locations, remote as the Polar Regions, background or rural, and urban locations with an average range between 1.5

ngm-3 (GEM) and 1 pgm-3 (GOM and PBM), depending on the speciation" should be revised: probably just by replacing "between" with "of".

Conclusion Obviously, an advanced comprehensive research has been done for the novel sensor design and manufacturing technology development. However, the sensitivity is far not sufficient to achieve the declared goal "to create promising sensors for environmental monitoring in real time, characterised by high sensitivity to the analyte" (lines 123-124). Probably, the new sensors can be used for other applications involving measurement of high mercury concentrations, such as technological or mercury exposure control. I would suggest that the manuscript No acp-2016-1077, after suggested corrections, can be redirected for publication in a journal dedicated to sensor technologies, which will be more suitable for this kind of research than Special Issue "Global Mercury Observation System – Atmosphere" (GMOS-A). The proposed novel Hg sensor may be made more suitable for the use in the GMOS project upon its further development and improvement.

---

## Referee Comment (RC2) · Anonymous Referee #2 · 27 Feb 2017

Peer Review of doi:10.5194/acp-2016-1077
*A Smart Nanofibrous Material For Adsorbing And Real-Time Detecting Elemental Mercury In Air*
 by Antonella Macagnano, Viviana Perri, Emiliano Zampetti, Andrea Bearzotti , Fabrizio De Cesare, Francesca Sprovieri and Nicola Pirrone

It is good to see research work attempting to develop new sensors for atmospheric mercury measurement.  The scientific work done by the authors is worthy of publication, with major modifications, but more on that below.

The communication of scientific work should be held to a high standard, in my opinion.  This English manuscript is very difficult to follow because of the poor grammar, sentence structure and basic mistakes; this impacts the scientific accomplishments.  For example, the word *vapor* is replace by the word *fiber* throughout the document, including the references, suggesting it was translated by an automated program and never reviewed.  I would estimate there are at least 100 places that need to be rewritten using acceptable English (missing words, misspelled words, confusing sentence structure and/or bad grammar).  There are too many to correct.  One example is line 333 "Sensing measurements, i.e. current (or resistance) changes, were provided in continuous."  The poor English and mistakes should be fixed, or else the manuscript should not be published.

**General comments on sensor technology**
One of the justifications for developing the technology reported in this manuscript is the current system used in nearly all major national and international networks is complicated and costly, which limits the atmospheric ability to monitor mercury worldwide (line 86).  I suggest that the limitation is the ultra-low levels of ambient mercury in the atmosphere.  The typical background gaseous elemental mercury (GEM) level of 1.5 ng/m$^3$ is equivalent to 168 parts per quadrillion by volume (ppqv).  There is no other atmospheric compound being measured routinely, continuously and automatically at this ultra-low concentration. Furthermore, current sensors in development, and even the equipment now used widely, are limited because of the need to collect mercury on a surface, like gold, with interferences commonly a million to billion times higher in concentration.  Moreover, for routine long term monitoring the mercury collection surface and system must be able to perform with stability, precision, accuracy, frequent calibration and most of all robustness over long periods in a wide range of complex and changing environments (high altitude, urban sites, tropics, deserts and mobile research platforms).  In research or commercial applications of automated, continuous air measurement technology, one near constant is that *complexity and cost increase as detection limit decrease*.  The conductive sensor made with nanofiberous gold, describe in the manuscript was making measurements in the ppb range, at least 1 million times higher than what would be required to measure GEM in the background or urban air.

**Specific critical comments to improve the manuscript**

The use of "Smart" in the title seems to be over-reaching and may set expectations too high.

Based on the comments above, the abstract may also be suggesting way too much in the last sentence (line 25).  At most it is a hope, or goal, that the sensor will be low-cost, very stable, low power and so on, since there was no scientific evidence or otherwise to give the reader the expectation that the claims will come true.

The justification and need for a new sensor was articulated very well.  It may be useful to comment on the challenges and limitations every mercury scientists faces, due to the ultra-low part per quadrillion levels of mercury in the atmosphere.

Please define the basis for the units of ppt and ppb used in the manuscript.  Typically concentrations for mercury used in the literature are mass/volume at standard temperature and pressure (e.g. $ng/m^3$).  Most gas monitoring is reported in volume/volume (e.g. pptv).   There is a factor of about 10x between $ng/m^3$ and pptv, which affects the understanding of the measurements made.

Since $CH_3HgCH_3$ was correctly listed as a gaseous oxidized form of mercury, then lines 44-45 must be changed, since dimethyl mercury is volatile and much less soluble that inorganic oxidized mercury forms.

The literature references of other mercury sensors using some form of nano-gold capture and detection was a good contribution to the manuscript.  There was the suggestion that this sensor in comparison to other sensors has encouraging results (lines 319-310).  It would have been useful to make a comparison of this work to the other mercury sensor technology being developed, for example Localized Surface Plasmon Resonance (LSPR) and state why this work is better or not.  A table is recommended.  Also, using a gold film and measuring the conductance change has been around for a long time as a successful, low-cost commercial instrument (Arizona Instruments, Jerome J405), able to measure Hg levels much lower than reported in this manuscript.  How is the nanofiberous gold system an improvement over the Jerome J405?

The description of the nanofiberous manufacturing process appears to be rather complex.  It is fairly well known that producing reproducible and robust surfaces with gold coating is difficult.  While it was stated that the manufacturing process was reproducible (line 314), with references, there was no evidence provided that the reproducibility of the manufacturing, would in turn lead to a predictable response of the sensor, or when it may fail after repeated sample/heat cycles. It also appears that there were two different ways to make the nanofiberous material, with the second starting with electro-spinning and sol-gel techniques.   Please clarify these two points with supporting evidence or at the minimum, comment on them.

The proof of concept, to measure low ppb levels of mercury in zero air, presented in the data figures and tables and discussed in the text is fairly well done.  Some questions and comments do remain about the sensor such as:
- In Figure 7, when clean air is introduced, the current keeps changing, when it would seem that it should be constant with no new mercury being added.  Please comment
- The need to have active flow over the sensor and use a curve fitting algorithm, begs the question of how the sensor system will be able to maintain accurate results when the Hg concentration varies over the 10 minute or longer sampling periods.  For example, if the concentration was 20 ppb for 2 minutes and 8 ppb for 8 minutes, how would concentration be determined?  I assume there would need to be some extrapolation between known response curves for different concentrations?  Please comment.
- There was no evidence shown that the response curve is stable with time and repetitive heating cycles.  Results of these experiments would be extremely valuable.
- Since there is active flow over the sensor, not through the sensor, there should only be a fraction of the gaseous mercury that is adsorbed onto the gold sensor surface (uptake rate).

Will the uptake rate change with flow rate, temperature of the sensor, and/or age of the sensor surface?  Please comment?

- Since the goal and expectation, as written by the authors, is to measure true ambient level Hg concentrations of 200 ppqv or lower, please provide potential improvements and technical advances that would make this possible?   As I understand it, the primary way to get to lower concentrations is very long sample times, which creates a number of trade-offs, such as no drift in electronics or temperature over time, greater potential for surface poisoning by interferences and most obviously, much lower resolution, to name just a few.  It would be good to have the authors comment on the needed technical advances and their feasibility to reach the 200 ppqv level.

The conclusion seems good overall.  In line 385, the phrase "highly sensitive to $Hg^0$" seems to be a bit of an overstatement and should be reworded.  Further in line 385, there is a statement about being robust and resistant to solvents and VOCs in air, but there was no data to support it or that it will not behave just like any other gold surface used for mercury capture and detection.  It would be better to state "that extensive experiments will be needed to determine if the gold surface will be robust against contaminants and interferences common in actual ambient air."  In line 391, why is the nanofiberous gold not considered a "trap", since it must collect (trap) mercury over a known time and then is reheated to start a new cycle  – I recommend this to be modified.  The last line of the conclusion is the key point for future work and evaluation of the sensor, so it is good to see the recognition of the challenges ahead.

---

## Author Response (AR2)

Reviewer 1
General comments.

The task of the atmospheric mercury monitoring network developing is very important for understanding the scale of emission, regional and global transfer, deposition of this environmental pollutant. Existing system of background monitoring is based on AFS and AAS instrumental observations requiring high investments for set up of any new monitoring point. That's why creation of new low-cost, hands off measurement systems is an imperative topical issue. The manuscript presents the results of the development of a novel sensor for air mercury measurements based on original manufacturing technology. The title reflects the contents of the paper, the mainresults are outlined in the Abstract. Introduction gives a comprehensive, 3.5 pages, review of the atmospheric mercury, speciation, mercury transfer, existing publications on the mercury sensors development. Principal part of the manuscript is devoted to description of a new sensor design and manufacturing technology, study of the absorptivity of the new material exposed to mercury vapour. Sensitivity of the new sensors is far not sufficient for the task of mercury monitoring declared in Introduction (see Specific comments, 1). No data on mercury measurement in ambient air, selectivity, and possible interferences are presented.

Specific comments
1) Commonly, in regulatory and scientific literature, the weight concentration units are used (ng/m3, pg/m3) for atmospheric mercury and mercury speciation. The authors use these units in Introduction, but different units: ppm, ppb, ppt in parts describing sensors. It is not explained what these units are related to: volume, mass, or number of molecules?

1) Authors agree to the Reviewer comments: scientists involved in monitoring mercury in atmosphere are used to adopt mass/volume as units. On the other hands, in scientific literature scientists involved in developing and characterizing the features of sensors or more complex devices are also used to employ units as ppm, ppb, ppt. In fact the scientific literature reported in this manuscript and related to sensors for mercury detection, reports mercury concentration in part per billion/trillion/million etc. Since the authors are describing the parameters of the proposed sensor as potential subject for atmospheric monitoring, units as parts per....were preferred.

We suppose the volume units are used. It is inconvenient, because a comparison with conventional concentration units requires recalculation
to the normal condition (P, T).

1a) Authors agree to the Reviewer about the necessity to clarify which kind of ppb (volume, mass, number of molecule). As described within the manuscript, a permeation tube was used as Hg generator and Tekran instrument as analyzer of the $Hg^0$ concentration. Thus a gas-tight syringe picked up defined microliters of air containing Hg° delivered in the measuring chamber and then injected into the analyser. When concentration was expected extremely high (800 ppb), the gas withdrawal was carried out upon a further dilution with the gas carrier. Tekran reported the results in ng/m$^3$, thus the final concentration was calculated taking into account the final volume (the dilutions) (5L/min - x5min)  and then properly converted in ppm/ppb from the formula commonly used in environmental sampling and analysis ( $C(ppb_v)$=concentration ($\mu g\ m^{-3}$) x $V_{mol}$ (L/mol))/Molar Mass(gmol$^{-1}$)). This is a conventional and common method to calculate the Hg

concentration in laboratory, therefore was not described. Vice versa units as ppb will be marked with the subscript vol.

Besides, the "tiny" ppb values can create a false impression of enabling background Hg concentration measurements. For example, mercury concentration of 1 ng/m3 is about 0.1 ppt (vol). It turns the achieved "low"detection limit (LOD) and measurement range of "20 -100 ppb (LOD 1,5 ppb)" (see lines 23-24) to a quite high figures in conventional units: 200,000 – 1,000,000 ng/m3 (LOD = 15,000 ng/m3). For comparison, the concentration of saturated Hg vapour is 12,000,000 ng/m3 at 18 oC. Such sensitivity gives no possibility to monitor mercury in ambient air, as the LOD is 10,000 times larger than the average background mercury concentration of 1.5 ng/m3 (see line 75).

1b) Authors knew that the range in which measurements were carried out could be related only to very polluted scenarios: this concept has been reported within the manuscript and in the Conclusion paragraph. The concentration range was selected in order to describe some sensing performances such as the response time and its relationships with the concentration, and the calibration curve that allows to determine the range of concentration where the sensor responses have a linear curve shape (up to about 40 ppb). The limit of detection here reported has been measured taking into account the sensor response changes (shift of current after Hg exposure) after 10 min (response time was clearly reported). Thus it was the minimum value at which the analyte can be reliably detected in a 10 min-measure. It was calculated according to the literature as 3xStandard Deviation of the electrical signal (signal-to-noise ratio) multiplied by the response/concentration ratio (the slope of the response curve). In fact, in a previous work (Macagnano et al., submitted, Macagnano et al., 2015a ) the authors reported a higher sensor sensitivity, with the possibility to detect up to dozens ppt, despite of a longer time necessary to reveal the analyte at these concentrations, in air. In this work the chance to apply the sensor in polluted sites and in real time has been presented and described (r145-149). Depending on the strategy of sampling and the coverage of the fibres, the limit of detection could be improved, about 2 ppt when slowly flowed within the measuring chamber (1 h measurement). However, despite the high sensitivity of the chemosensors to Hg0 vapour, the responses appeared to be slow.
What the present manuscript would like stress?
It describes the properties of a versatile sensor (ease of preparation, low cost apparatus) conductive at room temperature (low power) and sensitive to mercury vapours: this feature is depending on the time of exposure, due to the time necessary to entrap Hg enough to be detected as current variation. No carrier gas are required (it works in air). Sensor is robust: titania and gold are chemical compounds considered among the most robust materials. Thermal drift effects are expected to be lowered (only a 3min-heating process for desorption). Depending on the strategy of sampling and fibres treatments, the limit of detection could be improved. Potentials to be industrially fabricated (low cost of the process and the raw materials, electrospinning technology is commonly used in companies involved in textile, filtering and biomedicine) and used for large area monitoring (low dimension, low consumption, no gas carrier, no skilled operators). Their features make them promising candidates for sensor platforms (wired or WiFi connected).

2) How the LOD value was determined and at what exposure time? There is no description
in the text. Different figures of LOD are presented in Abstract and Conclusion (compare lines 23-24 and 390-391).

2) Authors described the way they calculated LOD within previous answer

3) Lines 69- 70 "Since automated measurement methods of Hg require power, argon
gas, and significant operator training, they are difficult to apply: : :" Argon is not required
for AAS systems.

3) The Authors sentence collected the several drawbacks related to the conventional equipment
for mercury detection. Obviously these instruments are extremely performing for mercury
detection, and they generally need gold traps to work. The mercury analysers based on gold
amalgamation and Atomic Absorption Spectrometry (AAS) detection are able to operate with
ambient air as carrier gas. One or two pure gold traps are installed in series to run the dual
amalgamation procedure and the sampling is run at about 1 L min-1 with sampling times of at
least 10 minutes. Under these conditions, a detection limit of about 0.1 ng m-3 is achieved. On the
other hand, some VOCs contained in ambient air may be adsorbed at the surface of the gold trap,
and then can cause interferences (broad bands) to UV light spectral absorption of mercury at
253.7 nm.

4) Lines 105-107 " decorated with gold nanoparticles (AuNPs) to obtain nanostructured materials
capable of adsorbing GEM as a useful alternative system for making regional and global estimates
of air Hg concentrations". Gold adsorbs not only GEM.

4) The sensor is expected to be robust such as the commercial sensors based on metal-oxides are.
The sensor is composed of $TiO_2$ and gold, two chemical compounds considered among the most
robust materials since resistant to  common solvents and VOCs as well as microorganisms attacks.
Specifically  titanium dioxide is insoluble in water, hydrochloric acid, dilute sulfuric acid, and
organic solvents and almost insoluble in aqueous alkaline media. It could be dissolved slowly in
hydrofluoric acid and hot concentrated sulfuric acid. Additionally gold is the most non-reactive of
all metals, never reacts with oxygen (one of the most active elements)  and water, which means it
is not subjected to corrosion. Both the compounds are thermally stable also in extreme
environmental condition. Additionally, the sensor is designed to work at room temperature
(decreasing thermal drift effects are expected), being thermally treated only for a few minutes (up
to desorb mercury from AuNPs (450°C). On the other hand, gold metal reacts with chlorine or
bromine, iodine and it dissolves in *aqua regia* but doesn't react with aqueous bases. Therefore,
contaminated environments by halides and sulphides should affect the adsorption of mercury,
thus cartridges or traps or coatings (as suggested in literature) for these interfering compounds
should be used or gold nanoparticles should be properly functionalized (e.g. alkanethiols).

5) Lines 116-118 This is true that: "sampling rates (SRs) using the same passive samplers may
depend on environmental conditions and atmospheric chemistry at each site. Moreover, it has
been also highlighted that the performance of passive samplers may be influenced by
meteorological factors (e.g., T _C, RH, wind speed) therefore inducing bias for the result of passive
sampling (Plaisance et al.,. 2004; Sderstrm et al., 2004)." However, there is no evidence that the
developed samplers are independent of meteorological factors (e.g. humidity) and interference of
trace gases and aerosols, especially under long exposure time.

5) In a cocktail of other chemicals, this sensor has been designed as a pretty selective sensor, being able to greatly decrease the environmental disturbances allowing the investigator/manufacturer to design/fabricate easier strategies to prevent contaminations from environment (selective filtering systems or coatings). Among common potential contaminants authors investigated water vapour influence (%RH) reporting no-effects on the electrical signals **[Zampetti et al.,** Procedia Engineering 120 ( 2015 ) 422 – 426**]**, but obviously, experimental data concerning the effects of further interfering gas (as halides and sulphides), as well as other physical parameters will be subject of further investigation (as specified in Conclusions paragraph).

6) Lines 121-124 "In this work we describe an alternative approach adopted in the place of conventional ones demonstrating that the combination of gold affinity for Hg with a nanoscale sized framework of titania provided the chance to create promising sensors for environmental monitoring in real time, characterised by high sensitivity to the analyte." See note (1). Besides, the technology of real-time monitoring is not described.

6) Authors disagree with the Reviewer about this question since Figs. 7 and 10 are transient responses (i.e. real time electrical signal changes under defined mercury concentrations).

7) Lines 128-131. "The TGM/GEM sensor surface described here could be deployed in a global network such as GMOS; a permanent network of ground based monitoring sites and observations of Hg and/or related species on a global scale and with remote sensors would in fact be highly desirable". Again, see note (1). The achieved sensitivity is insufficient for background measurement at the level of few ng/m3 provided by GMOS stations, and for speciation study at the level of pg/m3. 8) Lines 333-338. For better understanding of sensing measurements: "The sensor was exposed to a flow of Hg0 336 in air with a concentration of 800 ppb for 1 min (Fig. 7, right)" In terms of standard units, we see that the sensor was exposed to almost saturated Hg vapour at 8 mg/m3. Thereby it is not clear how such high concentration was measured? Earlier (line 234), it was mentioned, that "The Hg0 concentration was checked by Tekran®2537A analyser". Tekran 2537 is not capable of measuring such high concentration.

7) About those questions, authors clarified in a previous comment the sampling methods.
Technical corrections
1) Numerous misprints: "fibers" instead of "vapour":
Lines 91-91: "cold-fiber atomic absorption spectroscopy : : :. cold-fiber atomic fluorescence spectroscopy"
Line 141: "Morphological, optical, electrical aspects and sensing measurements of
fibers of GEM in air"
Line 144: "adsorbing and removal Hg fiber"
Thanks to Reviewer: all the typos and misprints have been properly modified
The same mistakes in the Reference list:
Lines 412-414 "Ferrua, N., Cerutti, S., Salonia, J. A., Olsina, R. A. and Martinez,
L. D., 2007. On-line preconcentration and determination of mercury in
biological and environmental samples by cold fiber-atomic absorption spectrometry.
J. Hazard. Mater. 141 693–699" In the original Cold Vapour, see:
http://www.sciencedirect.com/science/journal/03043894/150
Lines 412-414 "Determination by Cold Fiber Atomic Absorption Spectroscopy.
Analytical Letters 39" In the original Cold Vapour, see:
http://www.tandfonline.com/doi/full/10.1080/00032710600622167

2) Lines 73 -76 The sentence "Previous research highlighted that Hg-concentration levels in air vary greatly across different environmental locations, remote as the Polar Regions, background or rural, and urban locations with an average range between 1.5

ngm-3 (GEM) and 1 pgm-3 (GOM and PBM), depending on the speciation" should be revised: probably just by replacing "between" with "of".

A proper revision about typos has been provided.

Conclusion.
Obviously, an advanced comprehensive research has been done for the novel sensor design and manufacturing technology development. However, the sensitivity is far not sufficient to achieve the declared goal "to create promising sensors for environmental monitoring in real time, characterised by high sensitivity to the analyte" (lines 123-124). Probably, the new sensors can be used for other applications involving measurement of high mercury concentrations, such as technological or mercury exposure control.
I would suggest that the manuscript No acp-2016-1077, after suggested corrections, can be redirected for publication in a journal dedicated to sensor technologies, which will be more suitable for this kind of research than Special Issue "Global Mercury Observation System – Atmosphere" (GMOS-A). The proposed novel Hg sensor may be made more suitable for the use in the GMOS project upon its further development and improvement.

9) The authors conclusions are related to the resulting and then promising features of the sensor to be used for global mercury monitoring (low cost fabrication, low power consumption, no gas carriers, no skilled operators, miniaturization, versatility and potentials to be improved). Sensors provided with high surface area and porosity are suitable for exceptional interaction with environment, which means that, with the appropriate functionalisation, they are able to react with their target substance with great sensitivity. Therefore a potential commercialisation is significant. As previously explained, preliminary results are here reported: further investigation is required in order to design a final set-up configuration (an integrated microheater for desorption, a micropump, a set of filtering membranes for interfering) as well as an improved geometry of the sensor and the Hg delivery system. In Authors opinion and as reported within the manuscript, the proposed sensor sounds as one of the most promising sensors for mercury monitoring.

Reviewer 2

Peer Review of doi:10.5194/acp-2016-1077

*A Smart Nanofibrous Material For Adsorbing And Real-Time Detecting Elemental Mercury In Air* by Antonella Macagnano, Viviana Perri, Emiliano Zampetti, Andrea Bearzotti , Fabrizio De Cesare, Francesca Sprovieri and Nicola Pirrone

It is good to see research work attempting to develop new sensors for atmospheric mercury measurement. The scientific work done by the authors is worthy of publication, with major modifications, but more on that below. The communication of scientific work should be held to a high standard, in my opinion. This English manuscript is very difficult to follow because of the poor grammar, sentence structure and basic mistakes; this impacts the scientific accomplishments. For

example, the word *vapor* is replace by the word *fiber* throughout the document, including the references, suggesting it was translated by an automated program and never reviewed. I would estimate there are at least 100 places that need to be rewritten using acceptable English (missing words, misspelled words, confusing sentence structure and/or bad grammar). There are too many to correct. One example is line 333 "Sensing measurements, i.e. current (or resistance) changes, were provided in continuous." The poor English and mistakes should be fixed, or else the manuscript should not be published.

Authors' Answers

Authors thank the Reviewer 2 for his comments and suggestions in order to improve the manuscripts. About the English quality, authors agree to the Reviewer, then they proceeded to carefully review the manuscript's language. That issue was partly caused by the kind of grammar language set on the several computers used to write manuscript: sometimes American and sometimes British English. The attempts to homogenize the languages failed by program errors and authors' oversights.

**General comments on sensor technology.**

One of the justifications for developing the technology reported in this manuscript is the current system used in nearly all major national and international networks is complicated and costly, which limits the atmospheric ability to monitor mercury worldwide (line 86). I suggest that the limitation is the ultra-low levels of ambient mercury in the atmosphere. The typical background gaseous elemental mercury (GEM) level of 1.5 ng/m3 is equivalent to 168 parts per quadrillion by volume (ppqv). There is no other atmospheric compound being measured routinely, continuously and automatically at this ultra-low concentration. Furthermore, current sensors in development, and even the equipment now used widely, are limited because of the need to collect mercury on a surface, like gold, with interferences commonly a million to billion times higher in concentration. Moreover, for routine long term monitoring the mercury collection surface and system must be able to perform with stability, precision, accuracy, frequent calibration and most of all robustness over long periods in a wide range of complex and changing environments (high altitude, urban sites, tropics, deserts and mobile research platforms). In research or commercial applications of automated, continuous air measurement technology, one near constant is that *complexity and cost increase as detection limit decrease*. The conductive sensor made with nanofiberous gold, describe in the manuscript was making measurements in the ppb range, at least 1 million times higher than what would be required to measure GEM in the background or urban air.

Authors' Answers

All the comments and suggestions of the Reviewer 2 have been taken in consideration and the manuscript has been revised accordingly (red types).

**Specific critical comments to improve the manuscript**
The use of "Smart" in the title seems to be over-reaching and may set expectations too high.

1) Based on the comments above, the abstract may also be suggesting way too much in the last sentence (line 25). At most it is a hope, or goal, that the sensor will be low-cost, very stable, low power and so on, since there was no scientific evidence or otherwise to give the reader the expectation that the claims will come true. The justification and need for a new sensor was articulated very well. It may be useful to comment on the challenges and limitations every mercury scientists faces, due to the ultra-low part per quadrillion levels of mercury in the atmosphere. Please define the basis for the units of ppt and ppb used in the manuscript. Typically concentrations for mercury used in the literature are mass/volume at standard temperature and pressure (e.g. ng/m3). Most gas monitoring is reported in volume/volume (e.g. pptv). There is a factor of about 10x between ng/m3 and pptv, which affects the understanding of the measurements made.

Authors' Answers

All the Reviewer 2 clarifications have been taken in consideration and the manuscript has been revised accordingly (red types).

2) Since CH3HgCH3 was correctly listed as a gaseous oxidized form of mercury, then lines 44-45 must be changed, since dimethyl mercury is volatile and much less soluble that inorganic oxidized mercury forms.

Authors' Answers

The sentence has been properly modified.

3) The literature references of other mercury sensors using some form of nano-gold capture and detection was a good contribution to the manuscript. There was the suggestion that this sensor in comparison to other sensors has encouraging results (lines 319-310). It would have been useful to make a comparison of this work to the other mercury sensor technology being developed, for example Localized Surface Plasmon Resonance (LSPR) and state why this work is better or not. A table is recommended. Also, using a gold film and measuring the conductance change has been around for a long time as a successful, low-cost commercial instrument (Arizona Instruments, Jerome J405), able to measure Hg levels much lower than reported in this manuscript. How is the nanofibrous gold system an improvement over the Jerome J405?

Authors' Answers

A comparison with literature nanogold sensors has been added within the manuscript, as well as with the low cost commercial instrument of Arizona based on gold thin film (red types). The proposed sensor (LOD:2 $ppt_v$) was able to achieve comparable features of the Arizona system (Macagnano et al, 2017) under different measuring set-up systems. Long time measurements, reported also by Jerome J405, looked overcome by the flow rate. Furthermore the proposed sensor can be improved since subjected to further modification in density and size of AuNP, used with different or multiple transduction systems simultaneously (optical properties of nano-gold

particles) and could be supported by spots of UV-light irradiation in order to enhance the titania contribution into Hg adsorption (the oxidized Hg).

4) The description of the nanofiberous manufacturing process appears to be rather complex. It is fairly well known that producing reproducible and robust surfaces with gold coating is difficult. While it was stated that the manufacturing process was reproducible (line 314), with references, there was no evidence provided that the reproducibility of the manufacturing, would in turn lead to a predictable response of the sensor, or when it may fail after repeated sample/heat cycles. It also appears that there were two different ways to make the nanofiberous material, with the second starting with electrospinning and sol-gel techniques. Please clarify these two points with supporting evidence or at the minimum, comment on them.

Authors' Answers

The submitted paper of Ref.  has been recently published (now is available on-line), thus the proposed manufacturing procedure and the results related to the fabrication and reproducibility can be visualized and used.  Over the last 3 years of research, several sensors were used to measure the same concentration of Hg, and different sensors were investigated for different measurements, reporting comparable performances. About electrospinning and sol-gel techniques, there has been a misunderstanding: the electrospinning technique comprises sol-gel technique, when it is used to generate metal-oxide fibers, due to the precursor adopted for ES solution and calcination procedure. In order to avoid further misunderstandings, such a sentence has been clarified.

5) The proof of concept, to measure low ppb levels of mercury in zero air, presented in the data figures and tables and discussed in the text is fairly well done. Some questions and comments do remain about the sensor such as:
• In Figure 7, when clean air is introduced, the current keeps changing, when it would seem that it should be constant with no new mercury being added. Please comment

Authors' Answers

Yes, that was probably due to the Hg still present in the line at the time of the Hg measurement stop. This effect  has been confirmed by more recent measurements, where it disappeared when clean air never passed through part of tubes used  too to deliver mercury to the measuring chamber.

6) The need to have active flow over the sensor and use a curve fitting algorithm, begs the question of how the sensor system will be able to maintain accurate results when the Hg concentration varies over the 10 minute or longer sampling periods. For example, if the concentration was 20 ppb for 2 minutes and 8 ppb for 8 minutes, how would concentration be determined? I assume there would need to be some extrapolation between known response curves for different concentrations? Please comment.

Authors' Answers

The question is that changing the flow rate also the kinetics change. This effect and the strong relationships between the flow rate and sensor responses have been hugely investigated in literature. In the present manuscript only a specific flow rate (50 sccm) has been employed in order to understand if there was a relationship between the response curve and the concentration of the analyte selecting a few minutes of delivery time. Thus, using a selected flow rate you should have different transient response slopes for $Hg^0$ 20 ppb and 8 ppb concentrations, respectively, regardless of final current shift after a 2-min or 8-min responses.

The major expectation is related to the chosen of the best flow rate, capable to improve response time and the LOD of the sensor.

7) There was no evidence shown that the response curve is stable with time and repetitive heating cycles. Results of these experiments would be extremely valuable.

Authors' Answers

At the state of art, authors measured the current values of the same sensor after several heating cycles in order to measure potential thermal degradation effects: no electrical degradation was reported after tens of measures. Under investigation are sensor morphology, drift and lifetime.

8) Since there is active flow over the sensor, not through the sensor, there should only be a fraction of the gaseous mercury that is adsorbed onto the gold sensor surface (uptake rate). Will the uptake rate change with flow rate, temperature of the sensor, and/or age of the sensor surface? Please comment?

Authors' Answers

Yes, it would be. Sensor is exposed to a flow which impinges perpendicularly onto the surface of the sensor and then flows out from an exit positioned at the bottom of the sensor (outflow side). Temperature, age and flow rate are key parameters of the sensor.

9) Since the goal and expectation, as written by the authors, is to measure true ambient level Hg concentrations of 200 ppqv or lower, please provide potential improvements and technical advances that would make this possible? As I understand it, the primary way to get to lower concentrations is very long sample times, which creates a number of trade-offs, such as no drift in electronics or temperature over time, greater potential for surface poisoning by interferences and most obviously, much lower resolution, to name just a few. It would be good to have the authors comment on the needed technical advances and their feasibility to reach the 200 ppqv level.

Authors' Answers

Future expectations and strategies of improvements have been described within the manuscripts (red types)

10) The conclusion seems good overall. In line 385, the phrase "highly sensitive to Hg0" seems to be a bit of an overstatement and should be reworded. Further in line 385, there is a statement about being robust and resistant to solvents and VOCs in air, but there was no data to support it or that it will not behave just like any other gold surface used for mercury capture and detection. It would be better to state "that extensive experiments will be needed to determine if the gold surface will be robust against contaminants and interferences common in actual ambient air."

Authors' Answers

Line 385 has been reworded! About the sensor robustness and resistance, they are concepts coming from literature related to the physical, chemical and mechanical properties of gold and titania. Obviously extensive experiments will be needed to determine if the gold surface will be robust against contaminants and interferences common in actual ambient air. On the other hand, commercial systems (such as Arizona's one) have been provided of scrubbers and filters to reduce the contamination of gold by other air pollutants .

11) In line 391, why is the nanofiberous gold not considered a "trap", since it must collect (trap) mercury over a known time and then is reheated to start a new cycle – I recommend this to be modified. The last line of the conclusion is the key point for future work and evaluation of the sensor, so it is good to see the recognition of the challenges ahead.

Authors' Answers

Line 391 has been clarified according to the Reviewer suggestion.
* * *
Revised Manuscript

[revised manuscript text omitted]